# SELECTING OUT-OF-DISTRIBUTION DETECTOR FOR MULTIPLE MODALITIES

## ABSTRACT

Out-of-distribution (OOD) robustness is a critical challenge for modern machine learning systems, particularly as they increasingly operate in multimodal settings involving inputs like video, audio, and sensor data. Currently, many OOD detectors have been proposed, each with different designs targeting various distribution shifts. A single OOD detector may not prevail across all the scenarios; therefore, how can we automatically select an ideal OOD detection model for different distribution shifts? Due to the inherent unsupervised nature of the OOD detection task, it is difficult to predict model performance and find a universally best model. Also, systematically comparing models on the new unseen data is costly or even impractical. To address this challenge, we introduce M3OOD, a meta-learning-based framework for OOD detector selection in multimodal settings. Meta learning offers a solution by learning from historical model behaviors, enabling rapid adaptation to new data distribution shifts with minimal supervision. Our approach combines multimodal embeddings with handcrafted meta-features that capture distributional and cross-modal characteristics to represent datasets. By leveraging historical performance across diverse multimodal benchmarks, M3OOD can recommend suitable detectors for a new data distribution shift. Experimental evaluation demonstrates that M3OOD consistently outperforms 10 competitive baselines across 12 test scenarios with minimal computational overhead.

## 1 INTRODUCTION

Out-of-distribution (OOD) detection aims to identify samples that differ markedly from the distribution of the training data. For multimodal machine learning (ML) systems handling diverse modality inputs like vision, text, and audio, this capability is essential for maintaining robustness (Yang et al., 2021), and is particularly important in high-risk domains such as autonomous driving (Li et al., 2024) and medical diagnostics (Ulmer et al., 2020). As ML systems are increasingly adopted in multimodal settings (Radford et al., 2021; Zhang et al., 2023), researchers have begun to explore specialized benchmarks and frameworks for multimodal OOD detection (Dong et al., 2024a; Li et al., 2025). However, although a broad range of OOD detection methods have been proposed, each tuned to capture particular characteristics of data distributions, there is still no systematic approach for selecting the most suitable OOD detector under multimodal settings. This challenge stems from the inherently unsupervised nature of the OOD detection, which makes predicting model performance and identifying a universally optimal model challenging. Given that each OOD detector is based on distinct assumptions and methodological choices, selecting a single model for all distributions is ineffective, and exhaustively training one per case is infeasible. This is even compounded by the fact that cross-modal alignment inconsistencies and modality-specific distribution shifts can cause methods that perform well on individual modalities to fail when modalities are combined, consistent with the no-free-lunch theorem (Wolpert & Macready, 1997). Furthermore, conducting systematic model comparisons on the new data distribution shifts can be prohibitively expensive or unfeasible. As a result, we require an automated framework that can identify the most appropriate OOD detector without expensive evaluations.

Our solution is to learn from history: leverage the performance of detectors on many past datasets and transfer that knowledge to guide selection on new unlabeled datasets. This naturally connects to the paradigm of meta-learning (Vanschoren, 2018), or learning to learn, where knowledge gained across tasks is used to generalize more effectively to unseen tasks. In automated machine learning,

meta-learning has already been applied to model selection, providing evidence that such strategies can work in practice Zhao et al. (2021; 2022); Jiang et al. (2024); Ding et al. (2024). Building on recent work in unimodal OOD detection (Qin et al., 2025), extending to multimodal inputs introduces additional challenges: adapting embeddings to handle temporal sequences and spatial information, maintaining efficiency as dataset size grows, and creating unified representations for heterogeneous data types. More critically, designing effective multimodal meta-features for model selection becomes more difficult due to the complex interactions between modalities and the lack of clear guidelines for capturing cross-modal relationships in feature representations. Differences in data representation, distributional characteristics, and detection behavior across modalities require a specialized approach for multimodal OOD detection model selection.

To address the aforementioned challenges, we present M3OOD, the *first* model selection approach for OOD detection in multimodal settings, based on meta-learning. We show that by combining multimodal model embeddings with handcrafted meta-features capturing distributional and modality-specific properties, our meta-learning based approach can unify representations for video and optical flow. Alterations to a single modality can be reflected in the meta-embeddings, enabling the model to adapt its detector selection accordingly. The central idea is that an OOD detector that performs well on previous datasets with similar properties is likely to generalize well to new datasets. During the meta-training phase, we evaluate a pool of OOD detection methods across a wide range of carefully curated datasets spanning different modalities, including videos and optical flows, to build a performance profile under varied conditions. When a new multimodal dataset is introduced, we utilize the knowledge accumulated from historical datasets to recommend an appropriate OOD detection method. This selection process is guided by estimating the similarity between the new dataset and those seen during meta-training. Our main contributions are:

- **First Multimodal OOD Detector Selection Framework**. To our best knowledge, we introduce the first meta-learning-based framework for zero-shot multimodal OOD detector selection.

- **Specialized Multimodal Embeddings**. We use multimodal features to measure OOD detection task similarity, enabling better detector selection by capturing cross-modal OOD properties.

- **Superior Performance**. M3OOD surpasses eleven model selection methods across twelve test data pairs, yielding statistically significant ranking improvements with efficient runtime.

- **Open-Source Release**. We release the code at https://anonymous.4open.science/r/M3OOD-5C68.

## 2 RELATED WORK

### 2.1 MULTIMODAL OOD DETECTION

Multimodal OOD detection has gained attention in recent work, particularly for vision-language architectures (Ming et al., 2022; Wang et al., 2023). Maximum Concept Matching (MCM) (Ming et al., 2022) leverages alignment between visual features and textual concept representations to generate OOD scores. CLIPN (Wang et al., 2023) enhances the CLIP architecture through contrasting prompt techniques that strengthen the distinction between ID and OOD data. Furthermore, comprehensive multimodal benchmarks incorporating video, optical flow, and audio modalities have been developed (Dong et al., 2024b; Li et al., 2025), which expose cross-modal inconsistencies in OOD prediction and emphasize the need for methods that robustly integrate diverse signals.

### 2.2 UNSUPERVISED MODEL SELECTION

A key challenge in OOD detection is that the nature of OOD samples and their distributions is unknown during training (Hendrycks et al., 2019; Liang et al., 2018), making model selection necessarily unsupervised (Lee et al., 2018b). In practice, detectors must generalize to unseen distributional shifts (Yang et al., 2021), which underscores the need for unsupervised selection strategies (Liu et al., 2020b). Prior work has examined ensemble-based selection (Xue et al., 2024), whereas we aim to identify a single optimal detector per dataset. Existing approaches often rely on heuristics, such as defaulting to popular detectors like MSP (Hendrycks & Gimpel, 2017) or ODIN (Liang et al., 2017), or using ID confidence scores as proxies, but these can be unreliable due to neural networks' overconfidence on OOD data. Another direction is similarity-based selection, where models

are chosen based on dataset resemblance or clustering, a strategy adapted from algorithm recommendation (Kadioglu et al., 2010; Nikolić et al., 2013; Xu et al., 2012; Misir & Sebag, 2017). We include such methods as baselines in our study.

## 2.3 DATA REPRESENTATION IN META-LEARNING

In meta-learning, effective data representation is essential for capturing dataset or task similarity, and embeddings serve as a key mechanism for this purpose. Traditionally, computational meta-features such as dataset statistics and model-independent properties have been widely used to represent data in meta-learning frameworks (Vanschoren, 2018). More recently, advanced learning-based representations which aim to learn embeddings from data directly have emerged, including methods like dataset2vec (Jomaa et al., 2021) and HyPer (Ding et al., 2024). In parallel, language model and multimodal embeddings have been increasingly employed to encode dataset characteristics, offering a semantic-rich alternative that supports deeper model understanding (Drori et al., 2019; Fang et al., 2024; Qin et al., 2025). To leverage the strengths of both approaches, we combine handcrafted meta-features that capture distributional and multimodality-specific characteristics with SlowFast-generated embeddings for comprehensive multimodal dataset representation.

## 3 METHODOLOGY

### 3.1 PRELIMINARIES AND OOD DETECTION

**OOD Detection** aims to identify test inputs that do not follow the training distribution. Given training data $\mathbf{X}_{\text{train}} \sim \mathcal{P}_{\text{in}}$ and test data $\mathbf{X}_{\text{test}}$ containing both in-distribution (ID) and OOD samples, the goal is to construct a detector $M$ that determines whether each $x_i \in \mathbf{X}_{\text{test}}$ originates from $\mathcal{P}_{\text{in}}$. The detector $M$ is typically derived from a classifier $G$ trained only on ID data, and evaluation measures how well $M$ distinguishes between ID and OOD samples. When inputs span multiple modalities, this framework extends to **multimodal OOD detection**.

### 3.1.1 MULTIMODAL OOD DETECTION

Each training sample $x_i \in \mathbf{X}_{train}$ contains $K$ distinct modalities, expressed as $x_i = \{x_i^k \mid k = 1, \ldots, K\}$. Information from all these modalities is integrated to generate the final prediction by taking the combined embeddings from all modalities and outputs a score $s$. $s$ may represent a probability, a confidence score, an energy value, or any other scalar used by the detector. Let $\psi(\cdot)$ be a feature extractor that maps an input $x_i$ to an embedding $E_i$, and let $h(\cdot)$ be a scoring function that maps this embedding to an output score $s$. Thus,

$$s = h\big(\psi(x_i)\big) = h([\psi(x_1), ..., \psi(x_K)]) = h([E_1, ..., E_K]),$$

A sample $x_i$ with score above the threshold $\eta$ is classified as ID; otherwise, it is classified as OOD.

### 3.1.2 PROBLEM STATEMENT

Given a new and previously unseen pair of datasets $D_{\text{new}}$, our objective is to choose the best candidate OOD detection model $M \in \mathcal{M}$ *without* conducting test-time model evaluations, where we have no ground truth labels $\mathbf{y}_{\text{test}}^{\text{new}}$ for evaluation. In this work, we adopt a meta-learning approach to transfer performance knowledge from previously encountered tasks to the new OOD detection setting. This strategy is especially beneficial in situations where model evaluation is impractical or costly due to the absence of ground truth labels or the need for quick deployment. The proposed meta-learner in M3OOD, relies on:

1. **Historical dataset pairs** $\mathcal{D}_{\text{train}} = \{D_1, \ldots, D_n\}$ with ground truth labels, where $D = \{\mathbf{X}_{\text{train}}, \mathbf{y}_{\text{train}}\}$ and $\mathbf{X} = [\mathbf{X}^1, \ldots, \mathbf{X}^K]$ denotes multimodal inputs.

2. **Candidate model set** $\mathcal{M} = \{M_1, \ldots, M_m\}$ of $m$ OOD detectors.

3. **Performance matrix** $\mathbf{P} \in \mathbb{R}^{n \times m}$, where $\mathbf{P}_{i,j}$ records the performance of model $M_j$ on dataset pair $D_i$.

## 3.2 M3OOD FRAMEWORK

Our method includes two stages: (*i*) an offline training stage, where a model is trained to capture how different OOD detection models perform across a set of historical datasets $\mathcal{D}_{\text{train}}$, and (*ii*) an online stage, where this prior information is used to select an appropriate model for a new test dataset $D_{\text{new}}$. Fig. 1 outlines the workflow and key elements of M3OOD, with the offline training phase shown at top and the online model selection stage shown at the bottom.

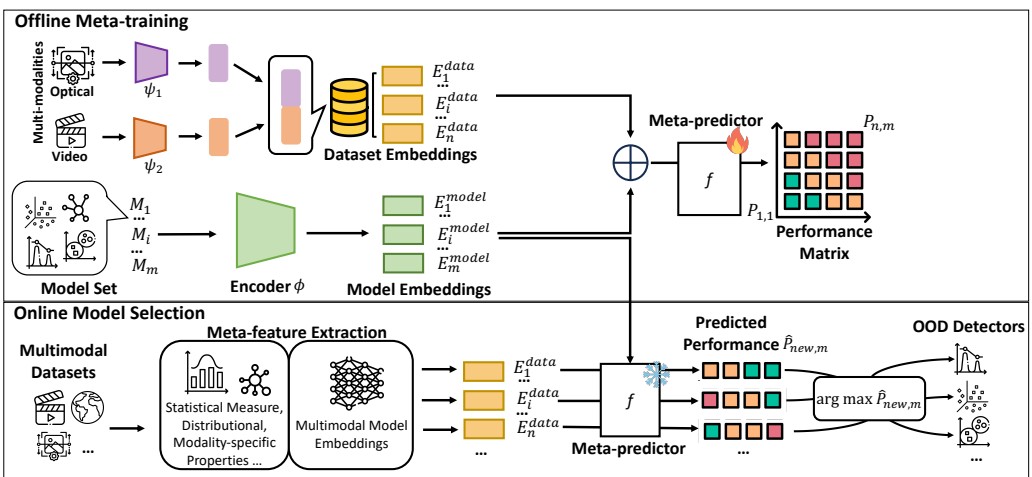

Figure 1: M3OOD workflow. The top part illustrates the offline meta-training phase, where the meta-predictor $f$ is trained. The bottom part shows the online meta-training phase, where the trained $f$ predicts performance on previously unseen multimodal datasets and selects the best model.

## 3.3 DATA AND MODEL EMBEDDINGS

A central component of our meta-learning framework involves extracting meta-features that characterize the essential properties of diverse datasets. Different OOD detection models employ varying algorithmic principles (e.g., probability-based, logit-based, feature-based) and operate under distinct assumptions about distribution shifts and OOD patterns. Consequently, model performance varies significantly based on the underlying dataset characteristics and types of samples present. When encountering a new task, the objective is to identify datasets in the meta-training repository that exhibit comparable properties and leverage models that have demonstrated strong performance on those similar tasks. This requires compact and consistent embeddings of data, allowing the meta-predictor $f$ to adapt effectively without depending on raw data of varying sizes.

The transition from unimodal to multimodal embeddings is driven by the limitations of single-modality representations in capturing cross-modal dependencies. While unimodal approaches are effective within their domains, they face representational bottlenecks and high-dimensional challenges. Multimodal fusion mitigates these issues by integrating complementary information, yielding more compact and discriminative representations. This is critical for OOD detection, where distributional shifts may appear differently across modalities. We therefore combine multimodal model embeddings with meta-features to improve task similarity assessment and detection performance, while representing OOD detectors with one-hot embeddings.

### 3.3.1 MULTIMODAL MODEL EMBEDDINGS

Multimodal models integrate information from multiple input sources by learning joint representations that capture cross-modal relationships and dependencies. In M3OOD, we use the SlowFast network (Feichtenhofer et al., 2019) initialized with pre-trained weights from Kinetics-400 (Kay et al., 2017) as our feature extractor to embed the visual information. For the optical flow encoder, we adopt the SlowFast architecture with only the slow pathway, using pre-trained weights from Kinetics-400 dataset (Kay et al., 2017). We concatenate the embeddings of all modalities and treat them as a unified entity.

### 3.3.2 TRADITIONAL META-FEATURES

Meta-learning fundamentally relies on transferring knowledge from previous tasks to improve performance on new ones, which is only effective when the new task shares structural similarities with historical tasks. The key challenge lies in defining robust representations of task similarity to identify which prior experiences are most relevant for the current problem. In meta-learning and feature engineering contexts, similarity between meta-train and test datasets are quantified through characteristic features of a dataset, also known as meta-features (Vanschoren, 2018). These features span from basic distributional characteristics like variance, skewness, covariance, etc. to multimodality descriptors that capture video temporal patterns and optical flow characteristics (e.g. colourfulness index, edge density, motion characteristics). Meta-features support task understanding by encoding both distributional properties, which are frequently employed in automated machine learning research, and modality-specific patterns, enabling model selection for new multimodal data through analogies with previously encountered tasks. The complete list of these features are in Appendix §2.

### 3.4 META-TRAINING

After the generation of the embeddings for each pair of training dataset $D_{\text{train}}$ and method $M$, during the offline training phase (Fig. 1, top), we fit a latent function that maps these embeddings to their observed performance values $P$. This supervised learning process allows the meta-learner to capture the relationship between dataset–method pairs and their outcomes, so that it can later generalize to unseen datasets and select the method predicted to perform best, according to the learned mapping $(D_{\text{train}}, \mathcal{M}) \mapsto P$.

We formulate the meta-predictor $f$ training process as a regression problem. The inputs to this meta-predictor are $E_i^{\text{meta}}$, the embedding of the $i$-th dataset pair, and $E_j^{\text{model}}$, the embedding of the $j$-th OOD detector. The dataset embedding for a dataset pair $D$ is defined as $E^{\text{data}} = \psi(D)$, while the method embedding derived from one-hot encoding. The meta-predictor $f$ is trained by mapping the meta-features to the corresponding performances $P_{i,j}$ associated with dataset pair $D_i$ and OOD detector $M_j$. Formally, the meta-train process can be formulated as:

$$
\begin{aligned}
f : \mathcal{H} \times \mathcal{G} &\to \mathbb{R}^+ \\
\text{where } \mathcal{H} &= \left\{ \left[ E_i^{\text{video}}, E_i^{\text{flow}} \right] \mid i \in \{1, \ldots, n\} \right\} \\
\mathcal{G} &= \left\{ E_j^{\text{model}} \mid j \in \{1, \ldots, m\} \right\} \\
f(E_i^{dataset}, E_j^{model}) &= \hat{P}_{i,j} \quad \forall (i,j) \in [n] \times [m]
\end{aligned}
\tag{1}
$$

Our goal is to train the meta-predictor $f^1$ to relate the characteristics of the datasets and the OOD detectors to their observed performance over historical dataset pairs. Specifically, the meta-train objective is as follows:

$$
\min_f \sum_{i=1}^n \sum_{j=1}^m \mathcal{L}(f(\psi(D_i), \phi(M_j)), P_{i,j})
\tag{2}
$$

, where $\mathcal{L}$ denotes the loss between the predicted and actual performance across dataset–detector pairs. This formulation serves as the training loss for optimizing the meta-predictor $f$.

### 3.5 META-TESTING

During meta-testing (Fig. 1 bottom), we generate embeddings for the test dataset pair $D_{\text{new}}$, reuse the precomputed embeddings of each model in $\mathcal{M}$, and use the meta performance predictor $f$ trained during the offline phase, to estimate the performance of different OOD detectors. The model with the highest predicted score is then selected, following the procedure in Eq. (3).

$$
M^* = \arg \max_{M_j \in \mathcal{M}} \widehat{\mathbf{P}}_{\text{new},j}, \quad \text{with} \quad \widehat{\mathbf{P}}_{\text{new},j} = f(E_{\text{new}}^{\text{dataset}}, E_j^{\text{model}}).
\tag{3}
$$

---

[1]The function $f$ may be instantiated with any regression model. In this work, we employ XGBoost (Chen & Guestrin, 2016) for its balance between simplicity and expressiveness, and its strong feature selection capability.

Given the new dataset pair, the trained predictor $f$ is used to estimate the relative performance of various OOD detectors. Based on the performance predictions, the method ranked highest is selected[2], as indicated in Eq. (3). Notably, this approach is *zero-shot*, meaning it does not involve any model training on the test data.

## 4 EXPERIMENTS

Our experiments address the following research questions (RQ): **RQ1**: How effective is the proposed M3OOD in unsupervised OOD detector selection compared to leading baselines? **RQ2**: How do different design choices affect the perofmance of M3OOD? **RQ3** : How much time overhead/saving does M3OOD introduce to multimodal OOD detection?

**Datasets**. In real-world settings, OOD data often differ from ID data not only in semantics but also in domain. To better reflect such situation, we design the dataset to include two types of distribution shifts: Far-OOD and Near-OOD. In the Far-OOD setting, we treat a full dataset as ID and use other datasets with related tasks but no overlapping categories as OOD. This introduces both semantic and domain shifts between ID and OOD samples. In the Near-OOD setting, we split the categories within a single dataset into two disjoint groups: one used as ID and the other as

| Category | OOD Detection Model |
|---|---|
| Probability-based | MSP (Hendrycks & Gimpel, 2017) |
| | GEN (Liu et al., 2023) |
| Logit-based | MaxLogit (Hendrycks et al., 2022) |
| | EnergyBased (Liu et al., 2020a) |
| Feature-based | Mahalanobis (Lee et al., 2018a) |
| | ViM (Wang et al., 2022) |
| | $k$NN (Cover & Hart, 1967) |
| Activation Pruning | ReAct (Sun et al., 2021) |
| | ASH (Djurisic et al., 2023) |

Table 1: OOD detectors considered for model selection.

OOD. In this case, ID and OOD samples share the same underlying distribution, differing only in semantics. We use five action recognition datasets (EPIC-Kitchens (Munro & Damen, 2020), HAC (Dong et al., 2023), HMDB51 (Kuehne et al., 2011), UCF101 (Soomro et al., 2012), and Kinetics-600 (Carreira et al., 2018)), The Near-OOD and Far-OOD setup details are in Appendix §4.

**Train-test Split**. In the meta-training stage, we split the train-test sets as shown in Tab. 2. Each row corresponds to a different meta-train/meta-test split. For example, in the first row, the meta-train set includes HMDB and Kinetics (Near-OOD and Far-OOD), and the corresponding meta-test set includes UCF and EPIC (Near-OOD). This setup ensures that the meta-predictor is trained on diverse OOD conditions and evaluated on unseen datasets, allowing us to assess its ability to generalize OOD detector selection across both semantic similarity and distributional shifts.

| Train | Test |
|---|---|
| HMDB, Kinetics, HMDB-Far-OOD, Kinetics-Far-OOD | UCF, EPIC |
| UCF, EPIC, Kinetics, Kinetics-Far-OOD | HMDB, HMDB-Far-OOD |
| HMDB, UCF, EPIC, HMDB-Far-OOD | Kinetics, Kinetics-Far-OOD |

Table 2: Meta-train train/test split (datasets without the "-Far-OOD" suffix are Near-OOD).

**Model set $\mathcal{M}$**. We construct $\mathcal{M}$ with 9 popular OOD detectors (Tab. 1) spanning different paradigms of detection strategies.

**Meta-predictor $f$** (see details in previous Offline Meta-Training section). In this work, we instantiate $f$ with XGBoost (Chen & Guestrin, 2016) for its balance of simplicity and expressiveness.

**Evaluation**. To evaluate the performance of M3OOD against the baselines, we compare the rank of performance of the OOD detector selected by each method among all candidates. We use Area Under the ROC Curve (AUC-ROC) as the evaluation metric[3], and visualize the results using boxplots and a rank diagram that reports the average rank across all dataset pairs. A rank of 1 indicates the best-performing selection, 11 is the worst (10 baselines plus M3OOD). To test the statistical

---

[2]Although selecting the top-$k$ methods for ensemble use is possible, this work focuses on top-1 selection.

[3]Other metrics can be used at interest.

significance, we apply the pairwise Wilcoxon rank-sum test across dataset pairs with a significance threshold of $p < 0.05$.

Table 3: Various OOD detectors' performances on Near-OOD and Far-OOD dataset pairs. We use Area Under the Receiver Operating Characteristic curve (AUROC) as the metrics. We highlight the selected OOD method for each dataset pair in the test set in bold.

| | Near-OOD | | | | Far-OOD | | | | | | | |
| ID Dataset | | | | | HMDB51 | | | | Kinetics-600 | | | |
| OOD Dataset | HMDB51 | UCF101 | EPIC | Kinetics | Kinetics | UCF | EPIC | HAC | HMDB | UCF | EPIC | HAC |
|---|---|---|---|---|---|---|---|---|---|---|---|---|
| MSP | 87.74 | 95.73 | 67.59 | 76.16 | 92.48 | 87.95 | 89.07 | 92.28 | 71.75 | 71.49 | 82.05 | 75.07 |
| Energy | 87.46 | 96.06 | 68.29 | 75.49 | 87.81 | 84.22 | 92.22 | 90.23 | 76.66 | 72.38 | 88.05 | 80.15 |
| MaxLogit | 87.75 | 96.02 | 68.29 | 75.98 | 90.34 | 87.91 | 91.88 | 91.99 | 78.43 | 73.97 | 84.90 | 80.30 |
| Mahalanobis | 85.28 | 97.14 | 42.99 | 35.83 | 90.34 | 87.91 | 91.88 | 91.99 | 78.84 | 74.33 | 82.69 | 79.51 |
| ReAct | 87.09 | 95.85 | 65.89 | 73.80 | 95.01 | 89.34 | 93.66 | 94.56 | 71.88 | 70.55 | 84.98 | 75.15 |
| ASH | 87.16 | 94.02 | 67.92 | 76.16 | 95.35 | 92.41 | 98.46 | 95.39 | 80.84 | 78.20 | 82.99 | 85.93 |
| GEN | 87.49 | 95.64 | **68.52** | 75.33 | 95.45 | **93.53** | 99.30 | 95.66 | 83.77 | 84.19 | 83.30 | 88.20 |
| KNN | **88.46** | 96.93 | 63.60 | 74.64 | 96.70 | 92.33 | 98.97 | 97.26 | 84.30 | 82.54 | 83.47 | 96.58 |
| VIM | 88.06 | **97.66** | 65.60 | **75.47** | **98.74** | 94.42 | **99.63** | **99.16** | **81.51** | **78.38** | **83.50** | **99.30** |

**Model Selection Baselines**. Following prior work on meta-learning for unsupervised model selection (Zhao et al., 2021; 2022; Jiang et al., 2024; Park et al., 2023), we choose baselines that fall into four categories, summarized in Tab. 4. Additional details of these model selection baselines are provided in Appendix §3.

**Hardware**. All models are implemented on the Multi-OOD codebase (Dong et al., 2024b) and run on a multi-NVIDIA RTX 6000 Ada workstation.

## 5 RESULTS

In Fig. 6, we report the distribution of the true ranks of the top-1 OOD detector selected by each model selection method across the test data pairs. Also, we include the overall average-rank diagram in Fig. 2, which displays the mean performance rank of the OOD detector selected by each algorithm. To compare two model selection al-gorithms (e.g., ours with a baseline), we apply the Wilcoxon rank-sum test to the ranks of the top-1 models selected by each method, as shown in Fig. 5. We summarize the main findings as below:

| Model Selection Baselines |
|---|
| **No model selection or random selection** |
| MSP (Hendrycks & Gimpel, 2017) |
| Mahalanobis (MD) (Lee et al., 2018b) |
| Mega Ensemble (ME) |
| Random Selection (Random) |
| **Simple meta-learners (non-optimization)** |
| Global Best (GB) |
| ISAC (Kadioglu et al., 2010) |
| ARGOSMART (AS) (Nikolić et al., 2013) |
| **Optimization-based meta-learners** |
| ALORS (Misir & Sebag, 2017) |
| NCF (He et al., 2017) |
| **Large language models as model selectors** |
| GPT-4o-mini (OpenAI et al., 2024) |

Table 4: Categories of OOD detector selection method baselines in this study.

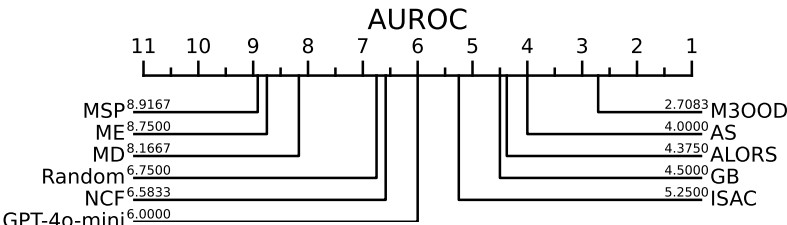

Figure 2: Average ranks of methods across datasets. M3OOD attains the lowest rank.

**1. M3OOD achieves superior performance compared to all baselines**. Fig. 6 demonstrates that M3OOD achieves stable, high-quality performance with minimal variance. It maintains the highest average ranking among all the 10 baseline methods that span from random or fixed selection to optimization or learning-based methods (Fig. 2). Additionally, Tab. 5 shows that most performance gains are statistically significant. This consistent pattern of results indicates that M3OOD effectively handles complex datasets while maintaining stable. We attribute this effectiveness to the integration of a meta-learning approach with our designed multimodal dataset embeddings.

**2. Meta-learner perform better than other baselines**. Meta-learners (M3OOD, AS, ALORS) significantly outperform single outlier detection methods and ME that averages all the model perfor-

| Ours | Baseline | p-value |
|---|---|---|
| | ME | 0.001 |
| | **AS** | **0.0625** |
| | ISAC | 0.0156 |
| | **ALORS** | **0.125** |
| **M3OOD** | Random | 0.0029 |
| | MSP | 0.001 |
| | MD | 0.0005 |
| | NCF | 0.0039 |
| | GB | 0.0312 |
| | GPT-4o mini | 0.0015 |

Table 5: Wilcoxon signed-rank test results (bold indicates no significance). M3OOD is statistically better than all baselines except AS and ALORS.

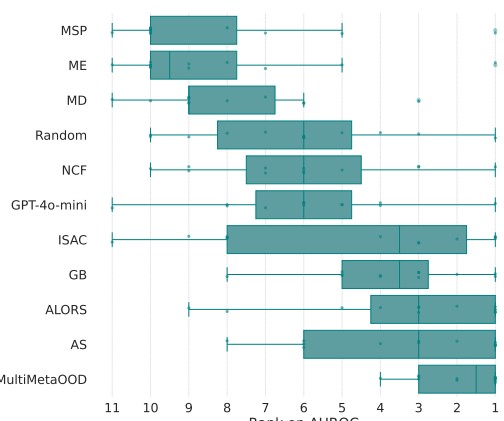

Table 6: Boxplot of the rank distribution on AUROC for M3OOD and the baselines. Rank 1 indicates the best performance, with lower ranks being better. M3OOD outperforms all baselines.

mances. Meanwhile, optimization-based meta learners (M3OOD, ALORS) demonstrate relatively strong and stable performance in model selection. This improvement comes from two factors. First, meta-learning uses knowledge gained from previous tasks to adapt more effectively to new ones, extracting shared patterns and representations that boost generalization. Second, the optimization routines in these methods drive models to efficient, high-quality solutions. With multimodal embeddings and meta-features, they predict model performance more accurately than simple meta-learners.

**3. The poor performance of the no-selection and random-selection baselines highlights the need for OOD model selection**. Simply averaging the OOD detection scores of all models yields subpar results, as shown in Fig. 2 and 6. Some models underperform consistently across datasets, so treating every model equally drags down overall effectiveness. While selective ensembles can help, building and running many models is often too costly. In contrast, M3OOD leverages offline meta-training to learn which single model to choose, avoiding ensemble construction and enabling efficient testing. Moreover, random selection falls short of all meta-learning baselines. This confirms that each meta-learner offers clear gains over random choice, and that picking an OOD detector at random is not advised. In addition, no single OOD detector achieves strong results on every dataset. This is because different OOD detectors target different dataset characteristics, and real-world data vary widely in their properties. Relying on just one method limits the range of solutions and makes it difficult to handle distribution shifts between datasets.

**4. LLM as zero-shot model-selector does not perform well under multimodal setting.** GPT-4o-mini may not be well-suited for capturing the nuanced relationships between multimodal dataset characteristics and OOD detector selection, which likely requires more specialized understanding of how different modalities interact and how various detectors respond to specific types of distribution shifts. This indicates that while LLMs offer accessibility, specialized meta-learning methods still hold substantial advantages for complex, heterogeneous-input settings such as choosing detectors for multimodal OOD detection.

## 5.1 ABLATION AND EXTENDED ANALYSIS

### 5.1.1 CHOICE OF META PREDICTOR

We evaluate the performances of different choice of meta-predictor $f$ and report the AUROC of the selected OOD detector for each dataset pairs. We test M3OOD with $f$ replaced by a two-layer MLP meta predictor. The result is shown in Fig. 3 left. Similar to what Jiang et al. (2024) observed in their study, XGBoost and other tree-based models produce more reliable and better-performing meta-predictors than neural network alternatives.

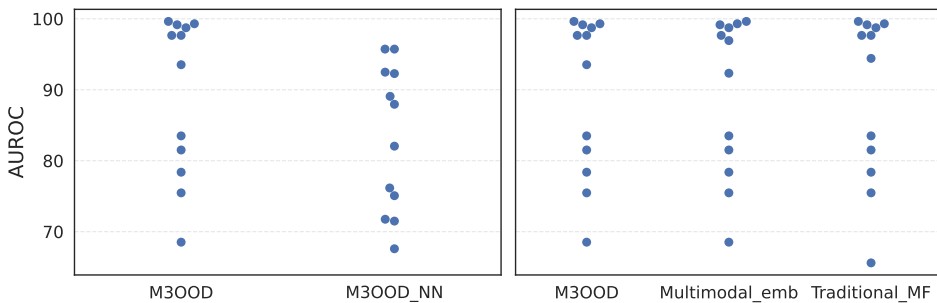

Figure 3: Left: ablation study on different choices of meta-predictor $f$. Tree-based models have better performance over neural network-based models. Right: ablation study on different meta embeddings. M3OOD has better performance over its variants.

### 5.1.2 CHOICE OF META EMBEDDING

We compare the performance of M3OOD altered meta-embedding inputs: one that excludes traditional meta-features (Multimoda_emb), and another that excludes multimodal model embeddings (Traditional_mf) across all the datasets. Fig. 3 right shows that combining multimodal model embeddings with traditional meta-features leads to improved performance. This suggests that the two types of meta-information provide complementary signals for selecting effective OOD detectors.

### 5.1.3 RUNTIME ANALYSIS

OOD detection on large datasets, especially in multimodal settings, is computationally expensive. Fig. 4 compares the runtime of M3OOD components against direct OOD detector execution on the HMDB dataset. While direct OOD detection requires extensive finetuning (HMDB requires 2 mins/ epoch for ~40 epochs, while Kinetics needs 10 mins/ epoch for ~40 epochs), M3OOD incurs minimal overhead with embedding generation (1527 seconds), meta-learning (57.8 seconds), and online selection (1.6 seconds) for the HMDB dataset. This demonstrates that M3OOD achieves significant computational savings compared to running OOD detectors directly.

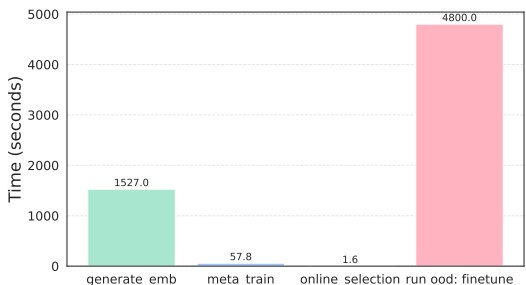

Figure 4: Runtime comparison of M3OOD components against execution time of multimodal OOD detection on the HMDB dataset. M3OOD adds small computational overhead relative to the overall detection process.

## 6 CONCLUSION

In this paper, we propose M3OOD, the *first* framework for multimodal OOD detector selection. Our meta-learner draws on a large set of historical OOD detector and dataset-pair records, using multimodal-related meta-features to guide model choice by learning from prior results. Extending embeddings from unimodal to multimodal enables the capture of cross-modal dependencies and leads to more effective multimodal OOD detection. However, M3OOD requires sufficient, high-quality historical dataset pairs, which can limit its performance when such data are scarce or not closely related. For future work, we will expand our evaluation suite to cover a broader range of datasets and models, thereby improving M3OOD's meta-learning capabilities. We also plan to integrate an uncertainty estimation component so that M3OOD can return an "I do not know" result when transferable meta-knowledge is insufficient, making it more reliable in challenging scenarios.

**Ethics Statement**: Our research follows the ICLR Code of Ethics, with attention to privacy, bias, and fairness. By enabling more accurate and unbiased model selection under multimodal settings, M3OOD reduces risks in sensitive domains such as surveillance and healthcare. Ongoing ethical evaluation ensures that M3OOD aligns with societal and legal standards.

**Reproducibility Statement**: We emphasize reproducibility in M3OOD. Detailed documentation of our methodology and experiments is provided in the paper and appendices. Our code are publicly available at `https://anonymous.4open.science/r/M3OOD-5C68` to support replication and further exploration.

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

## A    USE OF LLMs

LLMs are used to aid or polish writing. We used LLMs to assist with grammar and wording refinement of this paper. The research ideas, experiments, and analyses are solely by the authors.

## B    PSEUDO-CODE FOR META-TRAIN AND META-TEST

We discussed meta-training and online model selection in Section §3.3, 3.4 and § 3.5, respectively. Below are the pseudo-code for the two phases.

---

**Algorithm 1** Offline OOD detection meta-learner training

---

**Input:** Meta-train database $\mathcal{D}_{\text{train}}$ composed of $K$-modality data, model set $\mathcal{M}$
**Output:** Meta-learner $f$ for OOD detection model selection
1: Train and evaluate $\mathcal{M}$ on $\mathcal{D}_{\text{train}}$ to get performance matrix $\mathbf{P}$
2: **for** $i \in \{1, \ldots, n\}$ **do**
3:     Extract data embedding $E_i^{\text{meta}} = \psi(D_i) = [\psi(x_1), \ldots, \psi(x_K)]$
4:     **for** $j \in \{1, \ldots, m\}$ **do**
5:         Encode methods set as $E_j^{\text{model}} = \phi(\mathcal{M}, M_j)$
6:         Train $f$ by Eq. (1) with the $j$-th model on the $i$-th dataset
7:     **end for**
8: **end for**
9: **return** the meta-learner $f$

---

---

**Algorithm 2** Online OOD detection model selection

---

**Input:** the meta-learner $f$, New ID-OOD dataset pair $D_{\text{new}}$
**Output:** Selected model for $D_{\text{new}}$
1: Extract data embedding, $E_{\text{test}}^{\text{data}} := \psi(D_{\text{new}})$
2: **for** $j \in \{1, \ldots, m\}$ (for clarity, written as a for loop) **do**
3:     Encode methods set as $E_j^{\text{model}} = \phi(\mathcal{M}, M_j)$
4:     Predict the $j$-th model performance by the meta-learner $f$, i.e., $\widehat{\mathbf{P}}_{\text{new},j} := f(E_{\text{new}}^{\text{data}}, E_j^{\text{model}})$
5: **end for**
6: **return** the model with the highest predicted perf. by Eq. (2)

---

## C    DETAILS OF META-FEATURES

We introduced the multimodal model embeddings and meta-features for capturing multimodal OOD data characteristics in Section §3.4. Table 7 lists the complete set of meta-features we construct. Part of the meta-features are based on (Yi et al., 2023; Vanschoren, 2018; Wang & Schmid, 2013)

## D    DETAILS OF BASELINES

Section §4 introduces the model selection baselines, which we choose based on prior work in meta-learning for unsupervised model selection (Zhao et al., 2021; 2022; Jiang et al., 2024; Park et al., 2023). These baselines are grouped into four categories, as shown in main Tab. 4. Further details are provided below:

**(a) No model selection or random selection**: always employs either the ensemble of all the models or the same single model, or randomly selects a model: *(1) Maximum Softmax Probability (MSP)* (Hendrycks & Gimpel, 2017), a popular OOD detector that uses the maximum softmax score of a neural network's logits as threshold to identify whether an input belongs to the distribution the network was trained on. *(2) Mahalanobis (MD)* (Lee et al., 2018b) computes the distance between a sample's features and class means using the Mahalanobis metric, treating lower distances as more likely to be in-distribution. *(3) Mega Ensemble (ME)* averages OOD scores from all the models for a given dataset without performing model selection but rather using *all* the models. *(4) Random Selection (Random)* selects a model at random from the set of available candidate detectors.

| Meta-feature | Definition |
|---|---|
| *Video and Optical Flow Related Meta-features* | |
| Clip length $T$ | Number of RGB frames |
| RGB height $H$ | Height of each RGB frame in pixels |
| RGB width $W$ | Width of each RGB frame in pixels |
| RGB aspect ratio | $H/W$ |
| Flow height $H'$ | Height of each optical-flow frame |
| Flow width $W'$ | Width of each optical-flow frame |
| Flow aspect ratio | $H'/W'$ |
| Colourfulness index | Hasler–Süsstrunk measure computed from all pixels |
| Edge density | Fraction of Canny edge pixels, averaged over time |
| GLCM entropy | Average grey-level co-occurrence entropy over frames |
| HoF histogram | Eight-bin, magnitude-weighted histogram of flow orientations covering $(-\pi, \pi]$ |
| *Basic Statistics* | |
| $\mu_I$ (Clip mean) | Mean value |
| $\sigma_I$ (Clip std) | Standard deviation value |
| $\text{skew}_I$ | Skewness of distribution |
| $\text{kurt}_I$ | Kurtosis of distribution |
| $\min_I$ | Minimum value |
| $\max_I$ | Maximum value |
| $\text{med}_I$ | Median value |
| $\text{IQR}_I$ | Interquartile range of intensities |
| $\text{Gini}_I$ | Gini coefficient of values |
| $\text{MAD}_I$ | Median absolute deviation value |
| $\text{AAD}_I$ | Mean absolute deviation value |
| $\text{CV}_I$ | Coefficient of variation (std/mean) |
| $p_{\text{out},I}^{1\%}$ | Proportion outside 1st–99th percentile |
| $p_{\text{out},I}^{3\sigma}$ | Proportion outside $\mu_I \pm 3\sigma_I$ |
| $\mu_M$ (Flow mean) | Mean of optical-flow magnitudes |
| $\sigma_M$ (Flow std) | Standard deviation of optical-flow magnitudes |
| $\text{IQR}_M$ | Interquartile range of flow magnitudes |
| $p_{\text{out},M}^{1\%}$ | Proportion outside 1st–99th percentile of flow magnitudes |
| $p_{\text{out},M}^{3\sigma}$ | Proportion outside $\mu_M \pm 3\sigma_M$ of flow magnitudes |

Table 7: Details of the meta-features. Meta-features include CLIP-based per-frame meta-features and the grayscale and motion meta-features.

**(b) Simple meta-learners** that do not involve optimization: *(5) Global Best (GB)* is the *simplest meta-learner* that selects the model with the largest average performance across all meta-train datasets. GB does *not* use any meta-features. *(6) ISAC* (Kadioglu et al., 2010) clusters the meta-train datasets based on meta-features. Given a new dataset pair, it identifies its closest cluster and selects the best-performing model in that cluster. *(7) ARGOSMART (AS)* (Nikolić et al., 2013) finds the closest meta-train dataset (1 nearest neighbor) to a given test datasetin terms of meta-feature distance, and selects the model with the best performance on the 1NN dataset.

**(c) Optimization-based meta-learners** which involves a learning process: *(8) ALORS*(Misir & Sebag, 2017) factorizes the performance matrix to extract latent factors and estimate the performance matrix as the dot product of the latent factors. A regressor maps meta-features to these latent factors. *(9) NCF* (He et al., 2017) replaces the dot product used in ALORS with a more general neural

architecture that predicts performance by combining the linearity of matrix factorization and non-linearity of deep neural networks.

**(d) Large language models (LLMs) as a model selector**: *(10) GPT-4o mini* (OpenAI et al., 2024) used as zero-shot meta-selector. The dataset and method descriptions are directly provided to the LLM, allowing it to select the methods based on these descriptions. Note there is no meta-learning here. Details are in Supplementary Material §3.

### D.1 GPT-4O-MINI

GPT-4o-mini (OpenAI et al., 2024) is used as one of the baselines, serving as a zero-shot meta-selector. The text inputs are as follows:

#### D.1.1 DATASETS DESCRIPTIONS

**EPIC-Kitchens**: A large-scale egocentric dataset collected by 32 participants recording daily kitchen activities in their homes. We use the subset from the Multimodal Domain Adaptation benchmark, comprising 4,871 clips from the 8 largest action classes in sequence P22: put, take, open, close, wash, cut, mix, and pour. Modalities include video, optical flow, and audio.

**HAC**: Contains 3,381 video clips of 7 actions: sleeping, watching TV, eating, drinking, swimming, running, and opening door, performed by humans, animals, and cartoon figures. Includes video, optical flow, and audio modalities.

**UCF101**: Comprises 13,320 YouTube video clips covering 101 actions with significant diversity in motion, appearance, and background. Modalities include video and optical flow.

**Kinetics-600**: A large-scale dataset of $\tilde{4}$80k 10-second clips spanning 600 actions. We use a subset of 229 classes (57,205 clips) to reduce class overlap with other datasets. Optical flow is extracted at 24 FPS using the TV-L1 algorithm, totaling 114,410 samples. Final modalities include video, audio, and optical flow.

#### D.1.2 OOD DETECTOR DESCRIPTIONS

**MSP**: Implements the Maximum Softmax Probability (MSP) Thresholding baseline for OOD detection.

**EnergyBased**: Calculates the negative energy for a vector of logits. This value is used as the outlier score.

**MaxLogit**: Implements the Max Logit Method for OOD Detection, as proposed in Scaling Out-of-Distribution Detection for Real-World Settings.

**Mahalanobis**: Calculates a class center for each class and a shared covariance matrix from the data. It also uses ODIN preprocessing.

**ReAct**: Clips the activations in some layer of the network (backbone) and forward propagates the result through the remainder of the model (head). In the paper, ReAct is applied to the penultimate layer of the network. The output of the network is then passed to an outlier detector that maps the output of the model to outlier scores.

**ASH**: Prunes the activations in some layer of the network (backbone) by removing a certain percentile of the highest activations. The remaining activations are modified, depending on the particular variant selected, and propagated through the remainder (head) of the network. Then uses the energy-based outlier score. This approach has been shown to increase OOD detection rates while maintaining ID accuracy.

**GEN**: Utilizes the entropy of the softmax output as a measure of confidence. In-distribution samples are expected to have higher confidence (lower entropy), while OOD samples will exhibit lower confidence (higher entropy).

**ViM**: Implements Virtual Logit Matching (ViM) from the paper ViM: Out-Of-Distribution Detection with Virtual-logit Matching.

**KNN**: Implements the detector from the paper Out-of-Distribution Detection with Deep Nearest Neighbors. Fits a nearest neighbor model to the IN samples and uses the distance from the nearest neighbor as the outlier score.

### D.1.3 PROMPT

The prompt provided to the LLM is structured as follows, with text descriptions of both the datasets and models provided. To ensure consistency, we set *temperature* parameter to 0, and *top_p* parameter to 0.999.

```
 [Dataset descriptions provided]

 Your task is to select the best OOD detection method for a set
of 24 test ID-OOD dataset pairs.  You will be provided with
descriptions of both the ID-OOD dataset pairs and the available
OOD detection methods.  You should pick the best model that has
the highest AUROC metric.  For each dataset pair, output the
recommended OOD detection method in the format:  'Recommended
Method:  [Recommended Method]'.

 [Model descriptions provided]
```

## E   NEAR-OOD AND FAR-OOD SETUP

M3OOD utilizes five video datasets comprising more than 85,000 video clips in total. These datasets differ in the number of classes, which range from 7 to 229, and in size, ranging from 3,000 to 57,000 clips. Video and optical flow are used as different types of modalities. Details of the five datasets are in § 3.1.1.

In the **Near-OOD** setup, four datasets are used. EPIC-Kitchens 4/4 is derived from the EPIC-Kitchens Domain Adaptation dataset (Munro & Damen, 2020), with four classes used for training as ID and four for testing as OOD, totaling 4,871 video clips. HMDB51 25/26 and UCF101 50/51 are constructed from HMDB51 (Kuehne et al., 2011) and UCF101 (Soomro et al., 2012), with 6,766 and 13,320 clips, respectively. Kinetics-600 129/100 uses 229 classes selected from Kinetics-600 (Carreira et al., 2018), each with about 250 clips (57,205 total); 129 classes are used as ID and 100 as OOD.

In the **Far-OOD** setup, HMDB51 and Kinetics-600 are used as ID datasets. For HMDB51 as ID, OOD datasets include UCF101, EPIC-Kitchens, HAC, and Kinetics-600. To avoid class overlap, we exclude 8 overlapping classes from HMDB51 (leaving 43 ID classes) and remove 31 overlapping classes from UCF101 (resulting in 70 OOD classes). For the remaining datasets, no overlap exists and their original classes are used. For Kinetics-600 as ID, the same OOD datasets are adopted. We use the same 229-class subset from the Near-OOD setup to reduce overlap. For UCF101, 11 overlapping classes are removed, leaving 90 classes as OOD. Other datasets are used as-is due to no class overlap.

## F   ADDITIONAL EXPERIMENT SETTING

We select the parameters for M3OOD and M3OOD_NN (used in the ablation study) through grid search. The final parameter configurations are provided in the code repository.

## G   ADDITIONAL RESULTS

Figure 5 shows the dataset embeddings visualization, with the embeddings reduced to 2D using t-SNE. We observe clear clustering patterns that reflect underlying similarities across datasets. For instance, datasets originating from the same source or sharing overlapping label spaces, such as Kinetics-HMDB and Kinetics-UCF, are located closely, indicating that the meta-features capture alignment in distribution or content. Similarly, HMDB-EPIC and HMDB-Kinetics are proximal to HMDB, suggesting consistency in the extracted features when paired with other datasets Moreover,

datasets involving HAC (e.g., Kinetics-HAC, HMDB-HAC) appear in a distinct region, separated from others. This spatial distinction implies that the HAC dataset exhibits different properties—such as lower visual diversity, temporal resolution, or action granularity—compared to datasets like Kinetics and EPIC. This separation also highlights the ability of the meta-features to reflect meaningful dataset differences relevant for model selection and generalization.

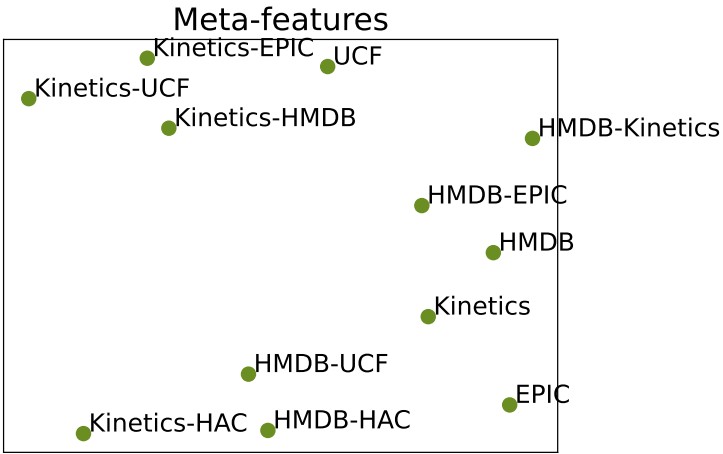

Figure 5: Visualization of dataset embeddings.

## H  DETAILS ON NOTATIONS

The following notations are used in Main Fig. 1 and Main Algorithm 2, which provides a comprehensive M3OOD overview.

| Notations | Description |
| --- | --- |
| $\mathcal{L}$ | Training Loss |
| $\mathcal{M}$ | # OOD Detection Methods |
| $\mathcal{N}$ | # Dataset Pairs |
| $D$ | Datast Pair |
| $\phi$ | Embedding Notation for OOD Detectors |
| $\psi$ | Embedding Notation for Dataset Pairs |
| $E$ | Embeddings for datasets and models |
| $P_{i,j}$ | Performance of OOD Detector $j$ on Dataset Pair $i$ |
| $\hat{P}_{i,j}$ | Predicted Performance of OOD Detector $j$ on Dataset Pair $i$ |
| $f$ | Meta-predictor |

Table 8: Notations with details used in Main Fig. 1 and Algo. 2.

