# OpenReview forum: "Selecting Out-of-Distribution Detector for Multiple Modalities"
_ICLR.cc/2026/Conference — ICLR 2026 Conference Withdrawn Submission_

### Official Review · Reviewer_i7wL · 2025-10-26

**Soundness:** 2
**Presentation:** 2
**Contribution:** 2
**Rating:** 4
**Confidence:** 3

**Summary:**

This paper presents M3OOD, a meta-learning framework for zero-shot selection of Out-of-Distribution detectors in multimodal settings. The approach leverages historical performance data and combines multimodal embeddings with handcrafted meta-features to recommend suitable detectors for new datasets without requiring expensive evaluations. Experiments on video and optical flow data show M3OOD achieves competitive performance against multiple baselines with minimal overhead.

**Strengths:**

1. The paper tackles the relevant challenge of automated OOD detector selection, extending it to multimodal contexts. The zero-shot selection paradigm is highly valuable for real-world deployment.
2. The runtime analysis convincingly demonstrates that M3OOD's overhead is negligible compared to the cost of running the OOD detectors themselves, highlighting its practical utility.

**Weaknesses:**

1. The Role of Table 3 is Unexplained and Misleading: Table 3, which presents the full performance matrix of all detectors on all dataset pairs, is the foundation of the meta-learning process and the primary evidence for the problem's existence. Yet, it is never mentioned or explained in the main text. This creates a critical disconnect for the reader.
2. The claim of a general "multimodal" framework is significantly undermined by the exclusive use of video and optical flow—two highly correlated modalities. This limited scope makes the contribution feel more like an incremental adaptation of established meta-learning model selection to a specific data pair, rather than a broader breakthrough for heterogeneous multimodal data (e.g., vision-language, audio-text).
3. Critical design choices lack thorough investigation. The justification for using XGBoost over more sophisticated neural meta-predictors is weak, and the simple concatenation for modality fusion ignores potentially more effective cross-modal interaction mechanisms.

**Questions:**

1. If I understand correctly, Table 3 is central to understanding both the problem and your method's training data. Why is it not discussed in the main text?
2. The performance of the Mahalanobis detector is catastrophic on some datasets (e.g., ~43% on EPIC). Does this indicate instability in the feature extraction or training process for certain detector-dataset pairs? Could such outliers negatively bias the meta-learner?
3. How does M3OOD's performance degrade when the test dataset is highly dissimilar from all meta-training datasets? Please provide an analysis of its sensitivity to the meta-training distribution.
4. Beyond concatenation, were more advanced fusion strategies for multimodal embeddings explored? Given the importance of cross-modal interactions, this seems a significant missed opportunity.

---

### Official Review · Reviewer_kdbY · 2025-10-29

**Soundness:** 2
**Presentation:** 2
**Contribution:** 2
**Rating:** 4
**Confidence:** 4

**Summary:**

This paper addresses the unsupervised model selection problem for out-of-distribution (OOD) detection in multimodal settings. The authors argue that since no single OOD detector universally excels across all data distributions, an automated method for selecting the best detector for a new, unseen task is necessary. They propose M3OOD, a meta-learning framework that learns from the historical performance of a pool of OOD detectors across a variety of multimodal datasets. The framework represents new tasks using a combination of learned multimodal embeddings and handcrafted meta-features.

**Strengths:**

1. The proposed problem is interesting
2. The empirical evaluation is comprehensive

**Weaknesses:**

1. The study evaluates only post-hoc OOD detection methods. In scenarios requiring strong robustness, the importance of selecting a single OOD detector may be overstated, as multiple detectors could be used jointly to enhance reliability.
2.  According to Table 3, the performance rankings of some OOD detectors vary only slightly across different datasets. This raises questions about the necessity of model selection when the performance differences are not substantial.
3. Using a meta-learning approach for OOD detector selection introduces biases from the meta-training data. This may limit the generalizability of the selected detector to new and dissimilar distribution shifts.
4. The runtime analysis in Figure 4 lacks clarity in its measurement setup. A more meaningful comparison should include total execution time, especially since M3OOD targets post-hoc methods. Inference time should also be compared to fully assess efficiency.

**Questions:**

N/A

---

### Official Review · Reviewer_PsuK · 2025-10-31

**Soundness:** 2
**Presentation:** 2
**Contribution:** 2
**Rating:** 2
**Confidence:** 4

**Summary:**

The paper proposes M3OOD, a meta-learning framework for selecting an OOD detector in multimodal settings. Experiments across Near-OOD and Far-OOD splits on several action-recognition datasets compare against 10 selection baselines

**Strengths:**

Generally clear and readable; figures and tables are informative; Codes are publicly available.

**Weaknesses:**

1. while this paper advertises multimodal selection across “video, audio, and sensor data,” the experiments fully focus on video. The applicability to text–image, audio, or sensor modalities is not demonstrated.
2. All benchmarks are action-recognition datasets with related statistics and similar backbones, which may not reflect broader multimodal OOD realities (e.g., text–image tasks with VLMs on imagenet-1k).
3. Common OOD metrics such as FPR@95% are absent.
4. Baseline set includes trivial strategies (Random, Global Best, Mega-Ensemble) and out-of-date methods (no baseline published on/after 2024 is included in this paper.)
5. Detectors are encoded via one-hot vectors, which restricts the practical usefulness since detector sets change frequently.

**Questions:**

see weakness

---

### Official Review · Reviewer_zP9D · 2025-11-01

**Soundness:** 3
**Presentation:** 3
**Contribution:** 2
**Rating:** 4
**Confidence:** 4

**Summary:**

This paper proposes M3OOD, a meta-learning framework for automated selection of out-of-distribution (OOD) detectors in video understanding tasks. The core idea is to train a meta-predictor that, given characteristics of a dataset and a pool of candidate OOD detectors, recommends the most suitable detector without requiring retraining or access to ground-truth OOD labels.  Evaluation is performed by comparing the rank of the selected detector’s performance (measured via AUC-ROC) among all candidates across multiple dataset pairs. Results show that M3OOD outperforms several baselines, including random selection, MSP, Mahalanobis, and ensemble methods, achieving higher average ranks and demonstrating generalization to unseen datasets.

**Strengths:**

1. The challenge of selecting an appropriate OOD detector without prior knowledge or extensive validation is highly relevant, especially in real-world deployment scenarios where manual tuning is infeasible. This work addresses a critical gap in practical OOD detection.

2. The construction of multimodal meta-features—including video-specific, optical flow, and statistical features—is thoughtful and domain-informed. These features go beyond generic data statistics and are tailored to video data, enhancing the plausibility of cross-dataset generalization.

3.  The paper employs a clean train-test split strategy across diverse datasets (HMDB, UCF, Kinetics, EPIC), ensuring that the meta-predictor is evaluated on unseen OOD conditions. The distinction between Near-OOD and Far-OOD settings adds depth to the evaluation.

**Weaknesses:**

1. The method is evaluated exclusively on video datasets. While the meta-features are well-designed for video, the paper overreaches in suggesting broad applicability to other modalities without empirical validation. Claims about "multimodal" applicability are misleading—the input data is multimodal (RGB + flow), but the framework itself is not tested beyond video.

2. There is no analysis of which meta-features or embedding components contribute most to performance. Without ablation studies, it is unclear whether the complexity of the complete feature set is justified or if simpler variants would suffice.

3. While XGBoost is a reasonable choice, the paper does not compare it to alternative meta-learners (e.g., neural networks, random forests) that might better exploit the structured meta-features or embeddings.

4. Relying solely on AUC-ROC rank as the evaluation metric obscures actual performance gaps. A method could be ranked highly but still underperform drastically in absolute terms. Reporting average AUC differences or failure rates would provide more insight.

**Questions:**

Can you quantify the contribution of different meta-feature groups (e.g., video-specific vs. statistical vs. model embeddings) through ablation studies? Is the performance gain worth the added complexity?

---

### Note · Authors · 2025-11-18

I have read and agree with the venue's withdrawal policy on behalf of myself and my co-authors.